# 3D Spheroids of Human Primary Urine-Derived Stem Cells in the Assessment of Drug-Induced Mitochondrial Toxicity

**DOI:** 10.3390/pharmaceutics14051042

**Published:** 2022-05-11

**Authors:** Huifen Ding, Kalyani Jambunathan, Guochun Jiang, David M. Margolis, Iris Leng, Michael Ihnat, Jian-Xing Ma, Jon Mirsalis, Yuanyuan Zhang

**Affiliations:** 1Wake Forest Institute for Regenerative Medicine, Wake Forest University Health Sciences, Winston-Salem, NC 27157, USA; faye@hospital.cqmu.edu.cn; 2Chongqing Key Laboratory of Oral Diseases and Biomedical Sciences, Chongqing Municipal Key Laboratory of Oral Biomedical Engineering of Higher Education, Stomatological Hospital of Chongqing Medical University, Chongqing 401147, China; 3SRI Biosciences, SRI International, 333 Ravenswood Avenue, Menlo Park, CA 94025, USA; kalyanij@gmail.com (K.J.); jon.mirsalis@sri.com (J.M.); 4University of North Carolina HIV Cure Center, University of North Carolina at Chapel Hill, Chapel Hill, NC 27599, USA; guochun_jiang@med.unc.edu (G.J.); dmargo@med.unc.edu (D.M.M.); 5Department of Biostatistics and Data Science, Division of Public Health Sciences, Wake Forest School of Medicine, Winston-Salem, NC 27101, USA; ileng@wakehealth.edu; 6Department of Pharmaceutical Sciences, The University of Oklahoma College of Pharmacy, Oklahoma City, OK 73117, USA; michael-ihnat@ouhsc.edu; 7Department of Biochemistry, Wake Forest University Health Sciences, Winston-Salem, NC 27101, USA; jianma@wakehealth.edu

**Keywords:** mitochondrial toxicity, 3D culture, spheroids, urine-derived stem cells

## Abstract

Mitochondrial toxicity (Mito-Tox) risk has increased due to the administration of several classes of drugs, particularly some life-long antiretroviral drugs for HIV^+^ individuals. However, no suitable in vitro assays are available to test long-term Mito-Tox (≥4 weeks). The goal of this study is to develop a 3D spheroid system of human primary urine-derived stem cells (USC) for the prediction of drug-induced delayed Mito-Tox. The cytotoxicity and Mito-Tox were assessed in 3D USC spheroids 4 weeks after treatment with antiretroviral drugs: zalcitabine (ddC; 0.1, 1 and 10 µM), tenofovir (TFV; 3, 30 and 300 µM) or Raltegravir (RAL; 2, 20 and 200 µM). Rotenone (RTNN, 10 µM) and 0.1% DMSO served as positive and negative controls. Despite only mild cytotoxicity, ddC significantly inhibited the expression of oxidative phosphorylation enzyme Complexes I, III, and IV; and RAL transiently reduced the level of Complex IV. A significant increase in caspase 3 and ROS/RNS level but a decrease in total ATP were observed in USC treated with ddC, TFV, RAL, and RTNN. Levels of mtDNA content and mitochondrial mass were decreased in ddC but minimally or not in TFV- and RAL-treated spheroids. Thus, 3D USC spheroid using antiretroviral drugs as a model offers an alternative platform to assess drug-induced late Mito-Tox.

## 1. Introduction

Antiretroviral therapy (ART) has significantly reduced HIV-related morbidity and mortality [1]. However, the therapeutic benefits of ART are often limited by delayed drug-associated toxicity [2,3]. As the life expectancy of HIV+ individuals has increased, the long-term safety of ART has gained increasing attention. Certain subclasses of ART induce mitochondrial impairment and cause mitochondrial toxicity (Mito-Tox), leading to the injury of the liver, muscle, heart, kidney, and central nervous system, even to death [4]. Mito-Tox has also been implicated in the post-market withdrawal of pharmaceuticals, yet mitochondrial impairment is difficult to detect in short-term drug screening studies. Therefore, it is highly important to develop suitable in vitro assays to identify the safety of antiretroviral regimens.

Conventional two-dimensional (2D) cultures of cell lines (i.e., HepG2 [5,6], or HepaRG [7,8,9]) have previously been reported for the evaluation of Mito-Tox, which serve as convenient tools for drug toxicity screening during drug development. These continuously growing cell lines are generally robust and easy to work with to quickly obtain basic toxicity information. However, cell lines are genetically altered, which can alter physiological properties and may not represent the in vivo state. Genetically altered cell lines can be further changed over passaging. In addition, traditional 2D cell cultures are not an accurate representation of how cells grow in vivo and how they are affected by disease and injury. Thus, short-term 2D cultures (≤2 weeks) of cell lines cannot always detect chronic toxicity occurring in clinical settings, nor can it predict the long-term Mito-Tox relevance of drugs. More than 90% of drugs that pass through in vitro 2D culture in preclinical studies fail to meet the desired efficacy or safety margins in the required subsequent clinical trials [10]. Further, there are only limited reports of Mito-Tox assays in 2D of cell lines or hollow fiber culture systems [11,12]. While long-term toxicity testing (≥4 weeks) is often performed in animal models [13], high rates of Mito-Tox observed in clinical trials with certain drugs suggest that animal toxicology studies may not be suitable for predicting Mito-Tox of ARTs intended for human use.

ART toxicity screening often requires human primary cells such as peripheral blood monocytes as a suitable in vitro culture system, however, these cells are not typically viable for long-term culture. Hence, there is an urgent need to develop more clinically relevant and predictive in vitro assays to assess compounds for the potential of late Mito-Tox. Growth in a 3D human primary cell culture would be a better way to represent the tissue ex vivo for Mito-Tox assessment. We are the first to demonstrate that stem/progenitor cells exist in human urine, i.e., urine-derived stem cells (USC) [14]. These cells can be obtained using a simple, non-invasive, and low-cost procedure. USC exhibit unique properties of adult stem cells, possessing a robust proliferative potential to generate sufficient cell numbers for regenerative applications [15,16,17,18,19,20,21,22,23,24,25,26,27,28]. We recently revealed that 3D cultures (i.e., spheres and organoids) afford a high viability rate of human USC for drug-induced nephrotoxicity during a 2-week culture [29], which offers a better microenvironment for cell growth, more closely mimicking in vivo conditions. Being easily accessible and mitochondria-rich cells [30], USC provide an optimal universal cell source in the assessment of Mito-Tox. With extended times for toxicity examination (shelf-life), mtDNA content increased 3-fold while the number of USC retained stably in 3D culture from weeks 2 to 4. Thus, in vitro 3D culture platforms seem to better correlate with chronic Mito-Tox under conditions of no cell proliferation with more limited mtDNA replication. Thus, the goal of this study is to develop an in vitro 3D culture system of human USC for ART-induced late Mito-Tox testing. We hypothesize that human USC maintain vigorous regenerative capacity and normal mitochondrial function in 3D organoids, extending the life of cell culture systems for ART-induced late Mito-Tox, which cannot be identified in 2D cultures.

In this study, we demonstrated that a 3D spheroid assay with human USC provides direct measurements of mitochondrial metabolism with high specificity and sensitivity in comparison with a 2D culture model system (Table 1). The nucleoside-analog reverse transcriptase inhibitor (NRTIs)\zalcitabine (ddC) [31] was used as an ART positive control due to its significant Mito-Tox in 2D culture and tenofovir (TFV) [32] was used as an ART negrative control due to its minimal Mito-Tox. In addition, raltegravir (RAL), an integrase strand transfer inhibitor (INSTI), is controversial in its ability to cause Mito-Tox [33,34] was used as a tested drug in these studies. We observed a transient decrease in Complex IV level induced by RAL following 2-week exposure. These observations suggest the 3D USC spheroids provide valuable information regarding Mito-Tox risk, enabling us to make informed decisions on drug development pipeline and personalized toxicology in clinical applications. As many newer INSTIs are unclear in their ability to cause Mito-Tox [33,34], this 3D system will be used in this regard for predicting in vivo Mito-Tox for new ART drugs by comparing our 3D model directly to animal data and to new clinical literature, when available. ART as a model system is used in this study, but the prediction of Mito-Tox would be useful for any drug class. We anticipate that the proposed in vitro long-term 3D culture of human primary cells will set the basis for future studies in predicting in vivo Mito-Tox for new ART drugs, other pharmaceuticals [4,5,35] (i.e., antiretroviral agents, anti-cancer drugs, certain antibiotics, anti-diabetic drugs, cholesterol lowering drugs, anti-depressants, and pain medications), environmental toxicants [36], industrial chemicals [37], and consumer products [38] as well.

## 2. Materials and Methods

### 2.1. Drugs, Chemicals, and Assay Kits

Drugs and chemical compounds: Zalcitabine (ddC) and Tenofovir disoproxil fumarate (TFV) were provided from the HIV Reagent Program; Raltegravir (RAL), Rotenone (RTNN), and Dimethylsulfoxide (DMSO) were purchased from Sigma-Aldrich (St. Louis, MO, USA). All drugs were dissolved at 1000 times at the final concentration in 0.1% DMSO and added 100 µL to 100 mL culture medium for treatments, with three biological replicates examined per concentration of each drug. Control cultures were treated with 0.1% DMSO.

Assay Kits: in vitro mitochondrial assay panel includes the following assay kits: NADH dehydrogenase (Complex 1) human simple step Enzyme Linked Immunosorbent Assay (ELISA) kit (ab178011, lot #GR3376581-1), CytC reductase (Complex 3) human profiling ELISA kit (ab124537, lot #GR3376911-1), Cytochrome c Oxidase (Complex 4) human enzyme activity assay kit (ab109910, lot #GR3330575-6), ATP assay kit (ab83355, lot #GR3364568-1) and caspase-3 assay kit (ab39401, lot #GR3362610-1), all purchased from Abcam (Cambridge, MA, USA) The Oxiselect™ total glutathione (GSSG/GSH) assay kit (STA-312, lot #10720210) and Oxiselect™ in vitro ROS/RNS assay kit (STA-347, lot #7081354) were purchased from Cell Biolabs Inc. (San Diego, CA, USA).

### 2.2. Isolation and Culture of USC

All human urine samples were approved for acquisition by the Wake Forest University Institutional Review Board (IRB00014033, approved on 17 February 2022). Informed consent was obtained from all subjects involved in the study.

Human USC were obtained from 8 healthy individuals aged 25–65 The mid- and last stream urine was collected and transported to the lab within 2 h. The urine samples from three individuals (at least three urine samples per donor) were mainly used as 3 biological independent replicates for all of the tests in this study. Urine samples from other 5 individuals were used as the backup in this study. USC were isolated by centrifuging the urine samples at 1200 rpm for 5 min. The cell pellets were washed twice by phosphate-buffered saline (PBS), resuspended, and plated on a 24 well plate. The modified culture media including keratinocyte-serum free medium (Gibco, Thermo Fisher Scientific, Waltham, MA, USA) and embryonic fibroblast medium (EFM), 5% fetal bovine serum (FBS), and other supplements purchased from Gibco mentioned in our previous works [14]. Briefly, Keratinocyte serum-free medium (Gibco #10724-011) was supplemented with 5 ng/mL epidermal growth factor (Gibco #37000-015), 50 ng/mL bovine pituitary extract (Gibco #37000-015), 30 ng/mL cholera toxin, 100 U/mL penicillin and 1 mg/mL streptomycin. Progenitor cell medium contained ¾ Dulbecco’s modified Eagle’s medium (Gibco #SH30243.01), ¼ Hamm’s F12 (Gibco #11765), 10% fetal bovine serum, 0.4 μg/mL hydrocortisone (Sigma H0888), 10^−10^ M cholera toxin (Sigma C8052), 5 ng/mL insulin (Sigma I6634), 1.8 × 10^−4^ M adenine (Sigma A2786), 5 μg/mL transferrin (Sigma T 1147) plus 2 × 10^−9^ M 3,39,5-triiodo-L-thyronine (Sigma T6397),10 ng/mL epidermal growth factor (Gibco #37000-015), and 1% penicillin-streptomycin. USC colonies (passage 0, p0) were found in 2D culture for 2 weeks. USC at p1 were cultured in 6 well plates, and sub-cultured in 10 cm dishes at p2. Culture medium was replaced every other day.

### 2.3. Preparation of 2D Culture and 3D Spheroids

USC at p3 were cultured in 96 well plates with an initial cell number of 8 × 10^3^ in 200 µL culture medium per well in the 2D culture at 96 wells (Thermo Fisher Science, Chicago, IL, USA). To establish 3D spheroid cultures, two types of ultralow attachment (ULA) well plates were used: (a) ULA 96 well plates (Corning, Glendale, AZ, USA) for USC plated in 200 µL culture medium with each well having 8 × 10^3^ cells to observe morphology of spheroids, measure cell growth and viability, mt-DNA content by quantitative real time-PCR (q-PCR), and mitochondrial mass by immunofluorescence staining, with the media changed every other day; (b) 81 spheroids (8 × 10^3^ cells/spheroid/well) were placed in a “81-well micro-mold plate” (MicroTissues 3D Petri Dish, Sigma, St. Louis, MO, USA); six “81-well micro-mold plates” were placed in a ULA 6-well-plate (Thermo Fisher Science, Chicago, IL, USA), which was convenient for daily changing medium. This “81-well micro-mold” was mainly used in testing cytotoxicity and Mito-Tox with enzyme-linked immunosorbent assay (ELISA) as this micro-mold is designed to generate sufficient cell numbers.; Total ATP content, Caspase-3, and total GSSG with Colorimetric assays, and ROS/RNS with fluorescence assay (Table 1 and Table 2). Drugs were added into the medium on the third day after seeding.

Each drug with 3 doses was added into a culture medium and incubated, Mito-Tox was evaluated in 3D spheroids, as compared to 2D cultures a three-time points (3 day, 2, and 4 weeks) Three FDA approved anti-HIV drugs were tested compared to two controls, including (1) ddC (0.1, 2, 10 µM) [39,40] known for significantly inducing Mito-Tox, 0.1 µM ddC is a typical single dose maximal plasma concentration (Cmax) [40]; (2) TFV (3, 30, 300 µM) [41,42,43] known for inducing only minimal Mito-Tox, 2.12µM TFV is a typical single dose maximal plasma concentration [40,41,42,43,44] (3) RAL (2, 20, 200 µM) [45,46,47] that is controversial in its ability to cause Mito-Tox, 2.25 µM is a typical RAL maximal plasma concentration [48], (4) rotenone (10 µM) [49] as a positive control; and (5) 0.1% DMSO as a negative control, see Table 1.

### 2.4. Cytotoxicity and Viability

Cytotoxicity was tested by cell count kit-8 (CCK-8, Dojindo, Japan) and MTT (Abcam, Cambridge, MA, USA) according to the instructions. Despite like MMT in cytotoxicity, CCK-8 assay requires washing spheroids after each test at different time points, which causes the risks in washing away of spheroids in 81-well micro-molds. Thus, CCK-8 was used in 2D culture while MTT was used in 3D spheroids. The medium of 2D cultured USC in multi-well plates (96 Well) (Thermo Fisher Science, Chicago, IL, USA) was changed to a fresh medium with 10% CCK-8 reagent. After incubation at 37 °C for 2 h, the medium of each well was carefully transferred to a new 96 well to read the absorbance at 450 nm.

At the same time point, the MTT assay was used to test USC in 3D spheroids. The spheroids in 81-well micro-molds were collected into one well to have enough cells for a reliable result (three replicates, n = 3 wells/sample). Briefly, all the spheroids in one mold were collected and incubated at 37 °C with an FBS-free medium and 50% MTT reagent for 3 h. The cultured media was discarded and MTT Solvent was added. The spheroids were treated with an Ultrasonic Cell Disrupter System to help completely dissolve the crystal in the cells. Briefly, spheroids with 300 µL MTT solvent in a 1.5 mL centrifuge tube were kept on ice. Each sample was sonicated with an ultrasonic probe with the intensity of 2 and 3–5 s per pulse for 5 pulses. The tube was centrifuged and 150 µL of the supernatant was taken to a 96 well plate for a plate reader. The absorbance was measured at 590 nm.

USC viability in 3D spheroid was examined with Live/Dead assay (Thermo Fisher Science, Waltham, MA, USA) by confocal microscopy. Calcein AM and EthD-1 were diluted by PBS into 2 mM and 4 mM as working solutions. The spheroid samples were washed with PBS, incubated by the working solution for 30 min. at room temperature (RT), and observed under a confocal microscope (Leica TCS-LSI, Leica Biosystems Inc. Buffalo Grove, IL, USA).

### 2.5. Hematoxylin and Eosin (H&E) Staining

Spheroids were washed once with PBS and fixed in 4% paraformaldehyde for 30 min at RT. Fixed cells were washed three times in PBS and embedded in HistoGel (Thermo Fisher, Chicago, IL, USA), and immersed in 70% EtOH for 2 days. Samples were run in a tissue processor (Tissue-Tek VIP 6 AI Tissue Processor, Sacramento, CA, USA) and embedded in paraffin. H&E staining of the sections was done by a Leica Histochemical Autostainer.

### 2.6. Fluorescence Staining for Mitochondrial Mass

MitoTracker Green FM is a green-fluorescent mitochondrial stain that appears to localize to mitochondria regardless of mitochondrial membrane potential (Thermo Fisher, Waltham, MA, USA). Mitochondrial mass was quantified by nuclear staining with Hoechst 33,342 (Thermo Fisher, Waltham, MA, USA). Cells were washed twice with PBS, incubated with 100 nM working solution at 37 °C for 15 min, washed again with PBS incubated with Hoechst 33,342 for 5 min, and then kept in fresh medium on ice before being observed by an Olympus FV10i confocal laser scanning microscope.

### 2.7. Quantitative Real-Time-PCR

Total DNA was purified from spheroids using a DNeasy Blood & Tissue Kit (Qiagen, Valencia, CA, USA) q-PCR reactions were run in duplicate using total DNA (5 ng/well), 200 nM forward and reverse primers, and the SYBR Green SuperMix (Thermo Fisher, Waltham, MA, USA) Quantitative q-PCR was performed to measure mtDNA content over nuclear DNA using the QuantStudio 3 Real-Time PCR System (Thermo Fisher, Waltham, MA, USA). The PCR temperature cycling conditions: 50 °C for 2 min, denature at 95 °C for 10 min, followed by 40 cycles of 15 s at 95 °C and 1 min at 60 °C, followed by 15 s at 95 °C, 1 min at 60 °C and a melting curve from 60 °C to 95 °C. Values were calculated by the following formulas: ΔCT = (nucDNA CT − mtDNA CT), Relative Mitochondrial DNA content = 2 × 2^ΔCT^.

Primer sequences are listed in Table 3.

### 2.8. Elisa

To obtain cell pellets, USC Spheroids treated with different drugs were rinsed with cold PBS (4 °C) twice within 1 min. The cell pellets were transferred to a 1.5 mL Eppendorf tube (VWR) and centrifuged at 300× *g* for 5 min. After the supernatant was discarded, the cell pellets immediately stored at −80 °C. For testing in the assays, the cell pellets were thawed on ice for 20 min at RT and then lysed using either the lysis buffer provided with Abcam Complex 4 assay kit or PBS buffer as per the protocol provided with the Abcam and cell biolabs assay kits. The total protein content of the lysates was then determined using the BCA assay. The levels of OXPHOS Complex I, III and IV in the cell lysates were determined using respective indirect ELISA assay kit (Abcam). The absorbance of this solution was measured at 450 nm on a Spectramax 190 (Molecular Devices LLC, San Jose, CA, USA) plate reader. The results for the assays were presented as % OXPHOS Complex levels which is calculated as described below: % OXPHOS Complex level = 100 × OD 450 nm measured in cell lysate for each concentration of test article collected at specific incubation time (day 3, week 2 or week 4)/OD 450 nm measured for the respective DMSO control collected at the same incubation time.

### 2.9. Colorimetric Assays

#### 2.9.1. Total ATP Content

The total ATP content of the cell lysates (1000 µg/mL per well) was determined by incubating the cell lysates with assay reaction mixture for 2 h, where the ATP in the lysates phosphorylated glycerol in the reaction mixture, generating a product that was quantified by measuring absorbance at 570 nm on a Spectramax 190 (Molecular Devices LLC, San Jose, CA, USA) plate reader. The % ATP level of the lysates was calculated as described below: % Total ATP level = 100 × OD 570 nm measured in cell lysate for each concentration of test article collected at specific incubation time (day 3, week 2 or week 4)/OD 570nm measured for the respective DMSO control collected at the same incubation time.

#### 2.9.2. Caspase 3 Content

The caspase 3 assay measured the activity of the enzyme in cell lysates by its ability to recognize and cleave the peptide substrate DEVD-p-NA. The cell lysates (1000 µg/mL per well) were incubated with the substrate DEVD-p-NA in a 96-well assay plate for 24 h. Wherein the substrate was cleaved by caspase 3 enzyme releasing the chromophore p-nitroaniline (p-NA), which was quantitated by measuring absorbance at 405 nm on a Spectramax 190 (Molecular Devices LLC, San Jose, CA, USA) plate reader. The % caspase 3 level of the lysates was calculated as described below: % caspase 3 level = 100 × OD 405nm measured in cell lysate for each concentration of test article collected at specific incubation time (day 3, week 2 or week 4)/OD 405 nm measured for the respective DMSO control collected at the same incubation time.

#### 2.9.3. Total Glutathione Content

The total glutathione content was determined by a colorimetric assay where the glutathione reductase enzyme reduced oxidized glutathione (GSSG) to reduced glutathione (GSH) in the presence of NADPH. Subsequently, the assay chromogen reacted with the thiol group of GSH to produce a colored compound that was monitored by measuring absorbance at 405 nm. The cell lysates (100 µg/mL per well) and assay standards in the 96-well plate was mixed with NADPH, glutathione reductase and chromogen solution provided in the assay kit and the color development (due to product formation) was monitored by measuring absorbance at 405 nm for 10 min with the reading taken every min on a Spectramax 190 (Molecular Devices LLC, San Jose, CA, USA) plate reader. The rate of color development is proportional to the concentration of glutathione in the cell lysate sample. The slope for each of the assay standards as well as the cell lysate samples was calculated by plotting the absorbance measured at 405 nm as a function of incubation time. The slopes calculated for the assay standards were plotted as a function of concentration and the total glutathione content in the cell lysates were calculated using the assay standard curve.

The results were presented as % total glutathione content and was calculated as follows: % Total Glutathione content = 100 × [GSSG] μM calculated from cell lysates for each concentration of test article collected at specific incubation times (day 3, week 2 or week 4)/[GSSG] μM calculated for the respective DMSO control collected at the same incubation time.

For both the ROS/RNS assay and Caspase 3 assay, we used cell lysates, rather than cells. We measured the total protein content of the cell lysates using BCA assay. Based on measured concentration, we loaded the same concentration of cell lysate in each assay well as per assay protocol. The OD means were normalized to the total protein of the cell lysates.

### 2.10. Fluorescence Assay

The ROS/RNS content of the cell lysates was determined by fluorescence assay carried out in 96-well black assay plates. The assay employed the fluorogenic dichlorodihydro-fluorescin DiOxyQ (DCFH) ROS/RNS specific probe, which reacted with ROS and RNS species in the cell lysate sample (1000 µg/mL) for 60 min to generate highly fluorescent DCF and measured at an excitation wavelength of 480 nm and emission wavelength of 530 nm on the Spectramax ID5 (Molecular Devices LLC, San Jose, CA, USA) plate reader. The fluorescence intensity (RFU) was directly proportional to the total ROS/RNS levels within the sample. The DCF content of the cell lysates was determined using the standard curve that was run simultaneously with the cell lysate samples. The results were summarized as % ROS/RNS levels with respect to the solvent-treated control collected at the same time point. % ROS/RNS content = 100 × [DCF] nM calculated from cell lysates for each concentration of test article collected at specific incubation time (day 3, week 2 or week 4)/[DCF] nM calculated for the respective DMSO control collected at the same incubation time.

### 2.11. Statistics

All data shown are derived from experiments that were independently repeated at least three times and analyzed by GraphPad Prism version 9.0.0 (GraphPad Software, San Diego, California, CA, USA). For experiments with multiple treatments, one-way ANOVA with Dunnett’s multiple comparisons to control group was used. For experiments with two factors (e.g., dose and time points), two-way ANOVA with Bonferroni’s multiple comparison test was performed. Statistical significance level was set at a 0.05. Statistical indicators represent statistically significant differences compared to the control groups. Data are presented as mean ± standard deviation (SD).

## 3. Results

### 3.1. Morphology, Cell Number and Viability of 3D Spheroids and 2D Culture of USC

To evaluate the stability of 3D USC spheroids, we measured the size, cell populations and viability at weeks 2, 4, 6 and 8. The sizes and shape-morphology of spheroids remained stable for 4 weeks of culture, but slightly decreased over time during culture before administration of ART. Renal tubular-like structures (arrows) formed and appeared at the optical surface of spheroids for 4 weeks, while the numbers of these tubular structures decreased over time. Live/Dead analysis confirmed that most cells (95%) survived (green) at week 4, however, about 10–20% of cells were dead (red) appearing at week 6 with a necrotic center (red) of spheroids formed at 8 weeks (Figure 1A,B). Thus, these results indicated that most cells in spheroids remained stable for at least 4 weeks, which is a suitable timing for ART-induced delayed Mito-Tox testing. Each of the generated results is summarized in Table 2.

In 2D culture, numbers of USC significantly increased at week 2 then decreased at week 4 compared those at day 3 in DMSO control in Figure 2. In addition, USC grew unstable after confluence, and the cells detached during 4-week culture in DMSO control before even being exposed to drugs (Figure 2C).

### 3.2. Cell Growth and Cytotoxicity in 3D Spheroids versus 2D Cultures after Drug Treatment

To evaluate the potential cytotoxic effect of antiretroviral drugs on 3D spheroids and 2D culture of USC, we measured the live/dead ratios of USC during 4-week cultures.

The spheroids remained stable in their size and shape (Figure 2A) in the treated groups with ddC, TFV and RAL for 4 weeks, which is similar to spheroids in the DMSO (0.1%) control group at 4 weeks. The non-proliferation status of USC remained stable (Figure 2B) 4 weeks after single ART drug administration. However, Rotenone (RTNN) (10 µM, positive control) significantly decreased the sizes of spheroids to around one-half of their initial size (Figure 2A) and showed significant cytotoxic effects with the cell death rate reaching about 40% (Figure 2B) and increases in red (dead) cells (Figure 2D) at 4 weeks. ddC treatment had mild cytotoxic effects while RAL, and TFV did not show cytotoxicity in USC in 3D spheroids up to 4 weeks. Approximately 98%, 95%, and 90% of cell survival rates were shown in 3D spheroids (Figure 2B,D) after the treatment of RAL, TFV and ddC, respectively, as compared to DMSO (99%) and RTNN (60%)-treated controls.

In contrast, USC in 2D culture proliferated rapidly and reached over confluence at week 1 (Figure 2A,C). USC continued proliferating after excessive confluence up to 4 weeks in the DMSO-treated control group. Although USC remained alive for 4 weeks, nearly half of the USC appeared to have cytoplasmic vacuolation, in particular in ddC-treated USC at low and middle dosages, and some cells appeared to have lysis in USC treated with ddC at high doses (Figure 2A) Vacuolization often accompanies cell death [50] after exposure to ddC. In addition, the number of cultured cells after confluence was unstable in 2D culture (Figure 2C) as cells often became detached and washed away in all drug-treated groups during the 4-week culture. In addition, sheets of cells detached during the measurement of cell population and immune-fluorescent staining procedures (data not shown). Thus, cell viability was not able to be accurately performed in 2D culture with Live/Dead analysis, particularly at later time points. Therefore, the following figures will only include data from the 3D USC spheroids due to the limitations mentioned above with the 2D culture.

### 3.3. Oxidative Phosphorylation System (OXPHOS) Complex I, III, IV

To determine mitochondrial function within 3D USC spheroids after ART, we measured OXPHOS Complexes I, III and IV. The control, RTNN, as expected, significantly inhibited the levels of OXPHOS Complex I at 2 and 4 weeks in a time-dependent manner (Figure 3A).

ddC at the mid and high dose (2 and 10 µM) significantly reduced the levels of OXPHOS Complex I and inhibited Complexes III and IV 4 weeks post treatment at 10 µM, as compared to DMSO control (*p* < 0.01) (Figure 3A–C). While small changes in the Complex III levels were observed at 2-weeks especially at the low ddC dose, a clear impact of ddC on the Complex levels was observed in USC cells following a 4-week exposure to ddC. This indicates the importance of long term (4 weeks) assessment of Mito-Tox in the in vitro models.

TFV did not result in a reduction in the levels of Complex I, II and IV levels at any dose or time point. RAL resulted in transient decreases in Complex IV at high doses at 2 weeks but not at week 4; nor did RAL have any impact on the Complexes I and III levels, compared to DMSO control (*p >* 0.05) (Figure 3). For all OXPHOS Complexes (I, II and IV) examined, basal (DMSO) levels of absorbance remained constant at 3 days and 2 and 4 weeks (Figure 3A–C, right-most graph), indicating the mitochondrial stability of the 3D USC spheroid system. Complex II and V in the USC cell lysate samples were not measured due to assay kit unavailability.

### 3.4. Total ATP Content Decreased after Single ART Drug Administration

As expected, RTNN significantly reduced ATP levels at day 3, week 2 and week 4 (38%; 66.4%; 27.4%, *p* < 0.01) respectively, assessed by colorimetric assays (Figure 4A). Significant increases in total ATP occurred 3 days after ddC treatment; while significant decreases in total ATP were observed in USC following treatment with all three ARTs: ddC at the high dose at week 2 and at all 3 doses at week 4; TFV at the middle or high doses at weeks 2 and 4; the newer drug RAL at all 3 doses at weeks 2 and 4. These drugs also resulted in a dose- and time-dependent reduction in total ATP from 2 to 4 weeks, indicating a delayed dose-dependent Mito-Tox. Total ATP levels remained consistent in DMSO control (Figure 4B).

### 3.5. mt-DNA Content and Mass in Treated 3D Spheroids

Quantitative real-time PCR analysis showed that ddC resulted in a dose- and time-dependent significant reduction of mtDNA (versus nuclear DNA) content, another marker of Mito-Tox (Figure 5). TFV at 30 µM and RAL at 2 µM significantly increased mtDNA content (*p* < 0.05) at 2 weeks. In contrast, the new agent RAL at the medium dose of 20 µM significantly reduced mtDNA content (*p* < 0.05) at 4 weeks (Figure 5A). However, mtDNA remained unchanged with TFV treatment, which is similar to that previously reported [51]. In addition, excessive mtDNA damage induced by RTNN displayed a compensatory increase in mtDNA content (i.e., mtDNA biogenesis) as has been noted in 2D culture [52]. mtDNA content significantly increased from week 2 to week 4 in 0.1% DMSO control treated USC in 3D spheroids (*p* < 0.01). (Figure 5B).

Mito-Tracker green, fluorescent staining at 4 weeks, an indication of mitochondrial mass, resulted in decreased (green) staining in the presence of ddC-treatment at 2 and 10 µM, while there were no significant changes in 3D spheroids treated with TFV and RAL (Figure 5C). This was associated with declined mtDNA content in USC treated with ddC alone (Figure 5A). Taken together, the data indicate that ddC decreased mtDNA content at week 2 and week 4, and reduced mitochondrial mass at week 4; RAL resulted in a decrease in mtDNA contend at week 4; and TFV resulted in little changes in any of these Mito-Tox markers.

### 3.6. Caspase 3 Activity

Significant increases in activity of the apoptotic marker caspase 3 were observed in 3D USC spheroids at 3 days and 2 weeks following treatment with all three drugs, particularly ddC (2 and 10 µM) at the early time point (day 3); TFV at the early and middle time points (day 3 and week 2); and RAL at week 2 (Figure 6A). This early caspase 3 induction may be due to localized hypoxia and other stresses after transfer of cells from 2D to 3D culture, as evidenced by higher levels of cleaved caspase 3 in the 3-day vs. 2- and 4-week cultures (Figure 6B) and once the cells acclimate to the 3D microenvironment, fewer cells have apoptosis which was confirmed in our Live/death kit study (Figure 2). Further, a similar phenomenon was observed in other studies in which cleaved-caspase 3 staining in rat liver spheroids showed some areas of apoptosis at early culture times; whereas the later treated samples showed negative staining of apoptosis at days 7, 11, 18, 21 [53]. As a control, RTNN induced significantly higher levels of caspase 3 at middle and late time points (weeks 2 and 4) (Figure 6B).

### 3.7. Measurement of ROS/RNS and Total GSSG

Fluorescent measurements showed that total ROS/RNS levels were significantly increased in ddC-TFV and RAL-treated UCS 3D spheroids at week 2 (*p* < 0.05), and low and mid-dose TFV and low dose RAL increased ROS/RNS but did not increase in ddC, TFV and RAL at high doses at week 4, as compared to DMSO control (Figure 7A). Interestingly, basal (DMSO control) levels of ROS/RNS increased with culture time, especially at 4 weeks, potentially explaining the relative lack of drug effects at this time point (Figure 7B). Finally, glutathione, a major intracellular antioxidant, was unchanged in 3D USC spheroids after any single ART drug treatment and time (Appendix A). This is presented in both reduced (glutathione) and oxidized (glutathione disulfide [GSSG]) forms (Appendix A).

The data represents by time in the regular Figure 3, Figure 4, Figure 6 and Figure 7 in the text are the same as those by drugs in the Appendix A.

## 4. Discussion

Highly active ART regimes have revolutionized the treatment of AIDS. Although ART successfully suppresses viral replication, significant toxicity after long-term ART can sometimes compromise treatment. Recognized toxicity that can be serious and irreversible is ART-related long-term Mito-Tox [54], which is associated with cardiomyopathy [55,56], myopathy [57,58], peripheral neuropathy [59,60], peripheral neuropathy [59,60], nephropathy [61,62,63] and hepatic steatosis with lactic acidosis [64,65,66], is often difficult to predict, and can be life-threatening [67]. Unfortunately, existing in vitro assays cannot reliably predict the likelihood that drugs might induce Mito-Tox. Thus, there is an urgent need to establish suitable in vitro models for the reliable prediction of Mito-Tox. The 3D USC spheroids are designed to be intermediate assays in toxicity assessment, to bridge the gap between traditional 2D cell culture and in vivo animal models [68], rather than replacing animal models of toxicity testing. The goal of this study is to develop and characterize an in vitro long-term 3D culture of human primary USC spheroid to reliably assess the impact of a single ART drug-induced delayed Mito-Tox for up to 4 weeks.

We generated an optimized assay of human primary cells for delayed Mito-Tox assessment using ART as a model. 3D spheroids of human USC possess several virtues. In the long-term culture in vitro (≥4 weeks) most USC remain alive and stable in 3D spheroids. This provides a long-term assessment of drug effects and is superior to 2D culture in which cells become confluent or non-adherent within 2 weeks. ddC at 10 µM significantly inhibited all OXPHOS Complexes I III and IV studies at week 4 (Figure 3), a finding not observed in other in vitro assays [69]. Further, 3D culture models more closely represent the stable cell numbers and mitochondrial content seen over time in human tissue in vivo. This allows a more accurate assessment of mitochondrial function, as compared to the rapid cell proliferation and DNA replication that occurs in 2D culture. The unpredictable nature of mitochondrial replication during rapid proliferation in 2D culture also makes it difficult to predict the likelihood of dysfunction [67]. Finally, the 3D spheroid system uses human primary cells that can be easily accessed, desirable for engineering human tissue equivalents in vitro, an advantage over the use of human cell lines or animal primary cells. Human USC have vigorous expansion capacity with large amounts of mitochondria and mitochondrial DNA (mtDNA) [70] and are highly sensitive to the effects on mtDNA and mitochondrial function. ddC has been documented to cause sensorineural deafness, hypertrophic cardiomyopathy, peripheral neuropathy [71,72], and liver toxicity [73]. In this study, ddC induced mitochondrial alterations, and reduced mitochondrial DNA (mtDNA) copy number in USC is comparable to what was previously observed in a ddC treated HepaRG cell line [73]. These observations indicate that despite its origin in the kidney, USC can potentially be utilized as a universal cell type to test drug-induced Mito-Tox, which is similar to the Mito-Tox tested in 3T3 fibroblasts [74], normal human skeletal muscle cells and RTEC [69].

The biological behaviors of USC are quite different between 2D and 3D cultures, as summarized in Table 4, which helps direct the optimal assay for Mito-Tox assessment. USC proliferate and reach confluence in the 2D culture on day 5 and continue to proliferate after cells reach over confluence at week 2. Then, cells start to detach when exposed to drugs. Thus, it is difficult to assay cell viability and challenging to obtain enough cells for a serial analysis after drug treatment due to detached cells grown in 2D because most (apparently dead) cells are detached and many washed away after each change of culture media. In addition, cell count viability measurements with MTT and live/dead analysis are consistent over time with 3D spheroids but not with 2D culture (Figure 1 and Figure 2). Furthermore, cells grow rapidly in 2D culture, and most ATP in cells growing in 2D is from glycolysis, but not OXPHOS routing [75], which is not suitable for Mito-Tox assessment. In contrast, in 3D spheroids, the number of cells and DNA replication are stable, which aids in the assessment of mitochondrial function, and mtDNA content. In addition, the dead cells in 3D spheroids can be easily detected and accounted for with Alive/Dead analysis as they remain within the spheroids (Figure 1).

USC reaching over-confluence without passaging led to cell detachment when cells were exposed to toxic drugs for 4 weeks. An alternative approach could be to subculture and passage cells before cells reach confluence and determine the cytotoxicity and Mito-Tox in 2D culture during 4-week drug toxicity testing. In this way, mtDNA copy number might increase with cell passages, which might affect the accuracy of results in Mito-Tox assessment. Further experiments could help to clarify this concern.

Morphologically, a 3D spheroid often consists of three cellular zones: (1) an optic zone with cell proliferation, (2) an intermittent zone with cellular quiescence and (3) a core zone with cellular necrosis [76,77]. The cellular necrosis in USC spheroids appeared at the core zone at week 6 and the size of the necrosis zone increased at week 8 (Figure 1), indicating that the optimal timeframe for toxicity testing is the first 4 weeks. It is important to determine the numbers of cells and timing to form healthy cell spheroids for cytotoxicity and Mito-Tox assessment. In the present study, our data showed that 4–8 × 10^3^ USC can generate spheroids keeping most cells alive and retaining a stable spheroid size stable at ≤350 µm for 4 weeks (Figure 1 and Figure 2). In addition, 3D-cultured cells show relative drug resistance as compared to 2D-cultured cells when forming dense 3D cells, and the resistance is associated with hypoxia [78]. Thus, 3D USC spheroids could be utilized as an in vitro cell-based drug-testing platform.

Functionally, in our negative control (i.e., DMSO)-treated spheroids, total ATP and GSSG remained unchanged from day 3 to week 4 (Figure 4 and Appendix A). The levels of apoptosis (Caspase-3) were much higher in 3D USC spheroid at day 3 compared to weeks 2 and 4 (Figure 6). This has been observed in another study when cells were transferred from 2D to 3D culture [53], which is attributed to the cells adapting to their new conditions. A ROS/RNS increase was also observed in other 3D spheroids, indicating the cells could be undergoing hypoxic conditions without cell apoptosis (caspase 3) and necrosis (live/dead) [79]. Thus, the size of spheroids should be kept to a maximum of 350 µm [80] so that oxygen and nutrition can be distributed into the inner portions of spheroids. In our positive control group, RTNN induced significant cytotoxicity, inhibition of Complex I, decrease in total ATP levels, and increase in caspase 3 expression in 3D USC spheroids at weeks 2 and 4, as expected. Taken together, these results indicate that cytotoxicity and Mito-Tox were induced in our 3D spheroid assay of human USC by RTNN, but not DMSO.

Drug-induced Mito-Tox has been documented to induce organ toxicity in several solid organs, including the liver, kidney, heart, central nervous system, and skeletal muscle. Drug classes include ART (NRTIs), anti-diabetic drugs (thiazolidinediones, fibrates, biguanides), cholesterol-lowering drugs (statins), anti-depressants (SARIs), pain medications (NSAIDs), certain antibiotics (fluoroquinolones, macrolides), and anti-cancer drugs (kinase inhibitors and anthracyclines) [81]. We used ART as a model system to detect of Mito-Tox in this study, which might be beneficial for drug screen and development. Thus, to use our 3D spheroid assay of USC for Mito-Tox assessment, we selected a well-known Mito-Tox associated NRTI ddC; a well-known NRTI agent tenofovir [82] with known minimal Mito-Tox, together with a largely unknown Mito-Tox INSTI drug RAL (Table 2).

Despite being potent in the treatment of HIV, ddC is associated with serious adverse events, including Mito-Tox and renal impairment. For these reasons, ddC is now rarely used in the clinics and has even been removed from pharmacies entirely [83]. ddC as a testing drug was selected to establish Mito-Tox in 3D spheroids of human USC in this study. Our data demonstrated that 4-week administration of ddC at each dose causes cytotoxic effects and leads to dose- and time-dependent mitochondrial dysfunction by inhibiting Complexes I, III and IV. ddC also inhibits mitochondrial replication by decreasing the expression of mtDNA content, decreasing total ATP levels, and inducing caspase-3 apoptosis and oxidative (ROS/RNS) stress in USC in 3D spheroids. These data indicate that a 4-week 3D assay of USC provides a viable model to detect later Mito-Tox that a traditional 2-week 2D assay might miss. In addition, ddC-treated USC display mtDNA depletion and significant reductions in mitochondrial DNA mass in our model (Figure 5). These reductions together are expected to affect the production of mtDNA encoded protein subunits of the electron transport chain, therefore, directly affecting oxidative phosphorylation (OXPHOS) (Figure 3) as has been found previously in Didanosine (ddI, NRTI)-induced-Mito-Tox [84].

As an NRTI, TFV has achieved therapeutic success for the treatment of HIV-1 infection. TFV has recently been reported with an increased risk of nephrotoxicity [85] despite no Mito-Tox induction in 2D cultures of HepG2 cells, human skeletal muscle cells, and renal proximal tubule epithelial cells [32], compared with other NRTIs. Another study showed that TFV achieves this independent of mtDNA depletion in HepG2 cell lines [51]. In this current study, we demonstrated that the administration of TFV for 4 weeks resulted in a significantly reduced production of ATP and a significant increase in both an apoptosis marker (caspase-3) and reactive oxygen species (ROS/RNS). However, TFV had no significant effect on cell growth, viability, mtDNA content, and the expression of Complex proteins of the electron transport chain (OXPHOS). As USC are renal cells, our findings suggest that TFV has a low potential to induce Mito-Tox in 3D USC spheroids, which is consistent with the results in 2- and 3-week monocultures of multiple cell types [32]. To determine whether Mito-Tox is the key target of TFV-associated renal tubulopathy, further investigation is needed. It has been shown that in some cases mitochondrial toxicity is not observed in media with glucose. Thus, the glucose/galactose approach will be used to reveal further toxicity (i.e., Crabtree effect [75]) in our 3D USC spheroid model in future studies. The Crabtree effect can be avoided by replacing glucose in the cell culture media with galactose, forcing cells to rely on oxidative phosphorylation to meet their energy demands due to galactose requiring two molecules of ATP to generate pyruvate and thus producing net ATP from anaerobic glycolysis [86].

INSTIs are a generally well-tolerated newer class of ART [87], and have a higher antiviral potency compared to other classes of ARTs. INSTIs include RAL, dolutegravir (DTG), bictegravir (BIC) and elvitegravir (EVG) [88]. Despite being controversial in causing Mito-Tox [33,34], INSTIs likely do impair mitochondrial function [89] by targeting the HIV integrase, which is responsible for integrating the reverse-transcribed viral DNA into the host DNA for further replication [90]. Despite a better safety profile of INSTIs compared to older NRTIs, there are concerns for the potential for long-term toxicity of INSTIs such as weight gain [91,92,93], cardiovascular effects, potential teratogenic effects (i.e., neural tube defects [94,95]), immune cell dysfunction [96] and neuropsychiatric manifestations [96,97,98], (i.e., insomnia [99,100,101]). The underlying mechanism of these adverse effects of INSTIs has not been completely identified. As an integrase strand transfer inhibitor (INSTI), raltegravir (RAL) is controversial in its ability to cause Mito-Tox [33,34]. A previous study showed that dolutegravir and elvitegravir (EVG), but not raltegravir (RAL), showed significantly decreased cellular respiration in a dose-dependent manner [96]. Another study reported that RAL did not induce Mito-Tox in the 2D culture of Hep3B cells and primary rat neurons at 24 and 48 h [102]. However, recent studies showed that raltegravir (RAL) and dolutegravir directly impacted adipocytes and adipose tissue by inducing oxidative stress, mitochondrial dysfunction, fibrosis, and insulin resistance [34]. Thus, RAL was selected as a drug for testing due to its unclear mitotoxic effect. In our ongoing studies, more ART drugs (such as dolutegravir, bictegravir, and elvitegravir as ISNTIs, islatravir as a nucleoside reverse transcriptase translocation inhibitor, and darunavir as a protease inhibitor) will be evaluated in this 3D system.

Our study is the first to show temporary Mito-Tox of RAL, specifically demonstrating a transient inhibition of OXPHOS Complex IV at week 2 while returning to control (DMSO)-treated levels at week 4 (Figure 3C), a reduction in total ATP (Figure 4) and an increase in apoptosis (caspase-3) (Figure 6) in 3D spheroids of USC for up to 4 weeks. In addition, long-term treatment with RAL was the only drug that led to statistically significant decreases in mtDNA content and mitochondrial mass (Figure 5). Thus, it is worthwhile to further study RAL and other INSTI drugs for their potential risk of inducing delayed Mito-Tox in long-term 3D cultures of human primary cells.

We dosed the spheroids every other day for up to 4 weeks, but it takes several years clinically for delayed Mito-Tox to manifest itself. Since ddC and RAL are typically dosed twice daily and TFV once daily, a patient would receive hundreds of doses of drugs before Mito-Tox is observed as compared to the maximal number 15 doses given in this experiment. Therefore, multiple doses around and above clinically achievable plasma levels (see Materials and Methods) were used.

Although several mitochondrial parameters were measured in this study, assessment of mitochondrial bioenergetic parameters (i.e., respiration and membrane potential) and glycolytic flux will further aid in our assessment of delayed Mito-Tox in future studies. Combining a 3D spheroid at a smaller size (≤200 µm) of human primary cells (such as neurocytes, hepatocytes, adipocytes or our USC) with Seahorse^TM^ technology [103] could more efficiently and accurately measure the real-time mitochondrial function of the tested USC, which could, in turn, improve personalized toxicology.

Overall, our 3D spheroid assay of USC has translational relevance for Mito-Tox and therefore may have potential in the assessment the inhibition of mitochondrial replication and function over prolonged exposure to other toxicants. Finally, this USC spheroid model could be used to test other types of delayed toxicity in addition to Mito-Tox. Preclinical drug development toward an Investigational New Drug (IND) Application requires subacute (typically two weeks) and more chronic (typically three months) testing in animals, and our model could provide an intermediate less expensive in vitro approach before proceeding to animals and perhaps to answer more mechanistic questions. Additionally, modeling chronic toxicity for environmental toxicants/mixtures in vitro has proven a serious challenge, and our USC spheroids could begin to provide such a model.

The development of a spheroid or organoid technique for toxicity testing in this study does have a few limitations, including (a) this technique can be costly; (b) spheroids lack in vivo tissue structures with multiple cell types and a vasculature; (c) spheroid systems provide limited numbers of cells (4 × 10^3^ cells/sample, n = 3) and Mito-Tox testing typically requires large numbers of cells (i.e., 5 × 10^6^ cells/sample, n = 3) for multiple assessments (i.e., ELISA, q-RCR, immunocytochemical staining, colorimetric assays, and Western blots) in long-term culture [11]; (d) the maximum of time periods for retaining most cell viability is in spheroid about 4–5 weeks. It would be beneficial to mai 3D spheroid culture longer to test chronic Mit-Tox. Recent studies showed new pharmacological and nonpharmacological approaches, such as hyaluronic acid hydrogels as natural CD44 targeting gels could improve long term cell retention of mitochondrial function in vitro [104]. It would be beneficial to use hyaluronic acid hydrogels or silk fiber matrices as we are developing to improve long-term cell viability in 3D culture for assessing chronic toxicities or reducing mitochondrial damage of antiviral therapies in future studies. Thus, a 3D culture system with new biomaterials might carry more cells stably at the level of the mitochondria for longer-term culture for Mito-Tox assessment. In addition, anti-diabetic drugs, such as Miglitol, can inhibit oxidative stress-induced apoptosis and mitochondrial ROS over-production in endothelial cells by enhancing AMP-activated protein kinase [105]. Thus our 3D spheroid model might be used in testing mitochondrial anti-diabetic drugs in prevention of mitochondrial side effects of ART and other agents. Furthermore, ART-induced Mito-Tox is caused by several mechanisms at play, including perturbations of cellular ribonucleotide and deoxy-ribonucleotide pools, which requires further investigation.

In summary, a novel long-term 3D culture assay of human primary USC has been successfully established for the evaluation of delayed Mito-Tox. Both RTNN and ddC, but not TAF and RAL, induced cytotoxic effects (Figure 2) and inhibited OXPHOS Complex levels (Figure 3), in comparison to control. Minimal changes in the OXPHOS Complex levels were observed following 4-week treatment with TFV and RAL (Figure 3), but there was a significant decrease in total ATP content of the USC following treatment with all four agents tested (Figure 4). An increase in caspase 3 activity was also observed in 3D USC spheroids at different time points following treatment with all four agents (Figure 6). An increase in ROS/RNS levels over DMSO control was also observed in 3D USC culture post-treatment with ddC and RTNN but not with TFV and RAL (Figure 7). No changes in USC total glutathione levels were observed post-treatment with any of the drugs (Appendix A). Our data demonstrated that 3D USC spheroids provide an in vitro assay for ART-induced long-term Mito-Tox and offer an advantage over previous 2D and hollow-fiber Mito-Tox assays.

## 5. Conclusions

In this study, we established a first-in-class approach using human primary urinary stem cell-based 3D cultures for assessing of Mito-Tox during drug development and for ultimately predicting Mito-Tox of long-term ART using patients’ own cells. This 3D spheroid technology could bridge gaps between in vitro and in vivo although it cannot completely replace animal systems in ART toxicology assessment. Such 3D spheroid systems with high-yield human USC could be used for initial higher throughput screening of drug candidates. As traditional ARTs, the classical NRTI zalcitabine (ddC) caused dose- and time-dependent Mito-Tox, while the NRTI tenofovir (TFV) induced minimal Mito-Tox in this long-term 3D culture spheroid platform. As one of the newer INSTIs, RAL caused transient Mito-Tox in our 3D spheroid assay. Increasing evidence has shown that other INSTIs might induce Mito-Tox, however, the causes remain controversial and the mechanisms are still not fully understood, which together require further study in a model like that developed here. In this study, we focused on a 3D spheroid assay for Mito-Tox induced by single ART drugs alone. Mito-Tox induced by effects of both HIV infection and multiple ART drugs could be studied with USC from HIV+ individuals because HIV infection itself, even during the latent HIV infection, is known to exacerbate Mito-Tox [106]. Furthermore, USC are optimal cells to develop 3D renal organoids for nephrotoxicity screening [29] as these renal stem cells can differentiate into renal tubule epithelial and other renal cells. Although a 4-week spheroid assay with USC from healthy donors may be superior to existing shorter toxicity assays, it is still unlikely to reveal long-term toxicity that develops following years of antiretroviral therapy. Thus, USC from HIV human subjects or individuals after drug treatment or pre-exposure prophylaxis could be used as an optimal cell source for establishing more personalized toxicological assays to test ART-induced delayed Mito-Tox. We predict that the use of HIV patients’ USC in 3D culture might reflect chronic MtT of years of ART.

## Figures and Tables

**Figure 1 pharmaceutics-14-01042-f001:**
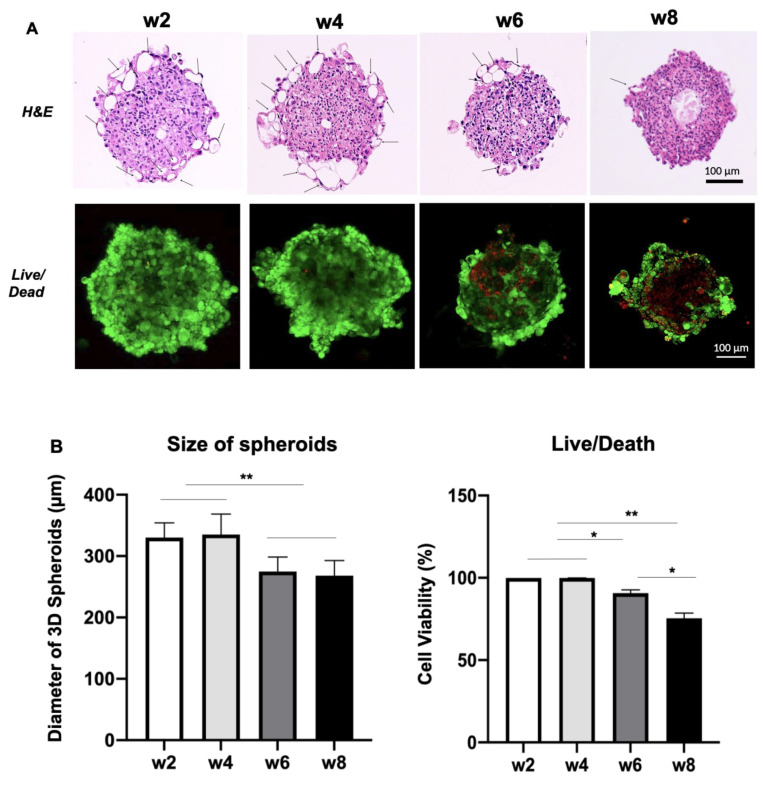
Changes in sizes and cell viability of 3D USC spheroids at different time points. (**A**) The sizes of spheroids remained stable at weeks 2 and 4 but decreased after 6 weeks. Renal tubular-like structures (arrows) formed and appeared at the optical layer of the spheroids. Most cells (95%) survived at 4 weeks, whereas about 10–20% of cells were dead (red) at 6 weeks. The initial seeding cell number is 4 × 10^3^ per well in 96 wells, assessed by H&E staining and live/dead kit on the tissue sections. (**B**) Quantification of size and cell viability of 3D spheroid as shown in (**A**). * *p* < 0.05; ** *p* < 0.01.

**Figure 2 pharmaceutics-14-01042-f002:**
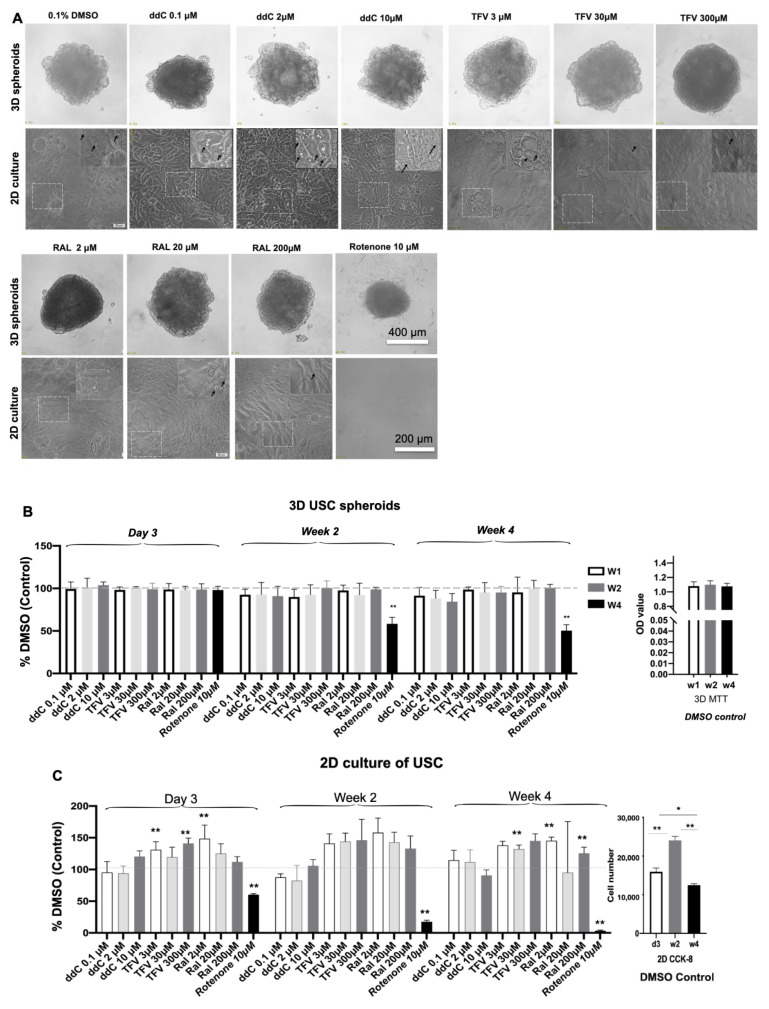
Changes in cell morphology, growth, and viability of 3D spheroids of USC 4 weeks after exposed to antiretroviral drugs. (**A**) Sizes of spheroids remained stable 4 weeks after ddC, TFV, RAL at different doses (low, medium, and high doses), which were similar to those in DMSO control, but significantly declined in the spheroid treated with RTNN, by phase contrast microscope (whole mount view) The vacuoles (dark short arrows) and cell lysis (dark long arrows) appeared in the cytoplasm of USC treated with ddC in 2D culture. Selected areas enlarged in inserts at the upright corner. (**B**) Numbers of USC remained stable with no cell proliferation in 3D culture in an ULA 81 well, by MTT assay. (**C**) Cell proliferation of USC in 2D culture in regular 96-wells, assayed by CCK-8. (**D**) Cell viability of USC in 3D spheroids, assessed with Live/dead staining, using confocal microscopy. USC at *p*4 were cultured at ULA 96 well. (8 × 10^3^ cells/well, USC at *p4*). * *p* < 0.05; ** *p* < 0.01.

**Figure 3 pharmaceutics-14-01042-f003:**
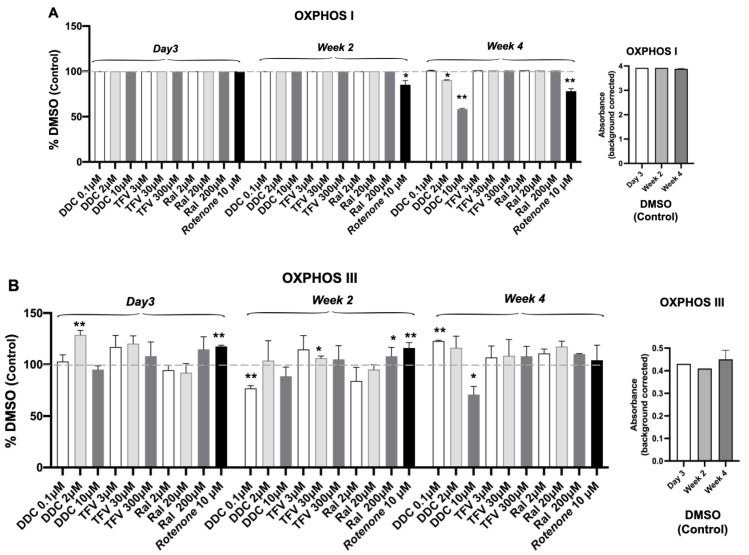
Effects of ARTs on Electron Transport Chain (ETC) OXPHOS Complexes. ELISA of subunits of electron transport chain (ETC) (**A**) Complexes I, (**B**). III and (**C**). IV in human USC in 3D spheroids 4 weeks after treated with ddC, TFV, and RAL at different doses (n = 3/sample/treatment/dose), DMSO (0.1%) and RTNN (10 μM) used as controls, respectively. * *p* < 0.05; ** *p* < 0.01.

**Figure 4 pharmaceutics-14-01042-f004:**
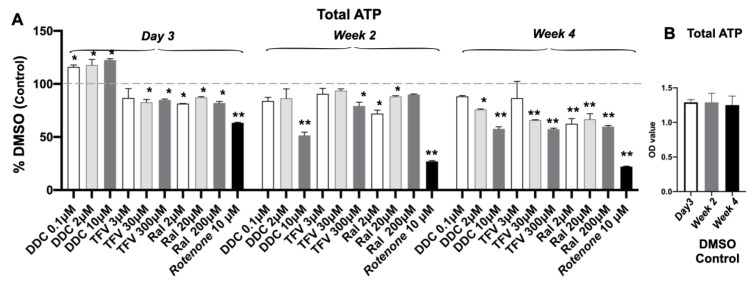
Total ATP content of USC in 3D spheroids decreased following treatment. (**A**). A decrease in total ATP over DMSO control was observed in USC spheroids following treatment with all three drugs at the highest dose tested at day 3, weeks 2, and 4. Particularly, the time-dependent Mito-Tox effect was observed in the treatment of ddC, TFV, RAL, and RTNN as control. (**B**). Total ATP levels were stable at DMSO control at the different points. The cell lysates of USC were assessed by respective colorimetric assays. Continued t-test *p*-values indicate statistical significance * *p* < 0.05; ** *p* < 0.01.

**Figure 5 pharmaceutics-14-01042-f005:**
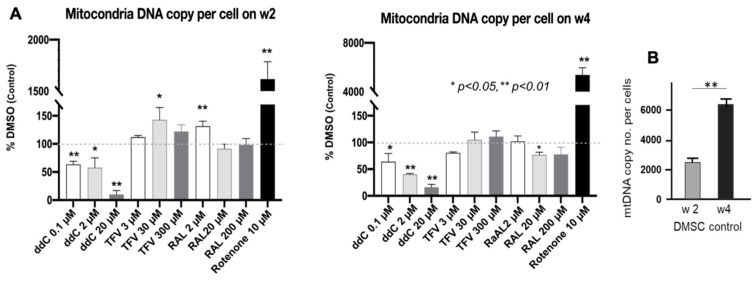
Chronic single ART drug exposure induced a decrease in mtDNA content and mtDNA mass of 3D spheres of USC. (**A**) Levels of mtDNA content significantly decreased in 3D USC spheroids treated with ddC, compared to control (DMSO) at weeks 2 and 4. TFV and RAL displayed slightly decreased mtDNA content at week 4, indicating the time-dependent effect of all three drugs compared to DMSO. Significantly higher levels of mtDNA contents were observed in 3D USC spheroids when exposed to RTNN (overcompensation), assessed by q-PCR (* *p* < 0.05, ** *p* < 0.01). Measurement of mtDNA content was done by normalizing the copy number of mtDNA to nuclear DNA (nDNA). (**B**). Levels of mtDNA content significantly increased at DMSO control from week 2 to week 4 (*p* < 0.001). A significant higher level of mtDNA was noted in 3D USC spheroids when exposed to RTNN (overcompensation), assessed by q-PCR (* *p* < 0.05, ** *p* < 0.01). (**C**). Mitochondrial mass of 3D USC-spheroids (*p*4) remarkably decreased 4 weeks after the treatment of all three drugs, ddC, TFV, and RAL, compared to 0.1% DMSO control; Particularly, ddC (2 µM and 10 µM) and TFV (300 µM) decreased the expression of mitochondrial mass with a dose-dependent effect, assessed as by Mito-Tox Tracker green fluorescent dye.

**Figure 6 pharmaceutics-14-01042-f006:**
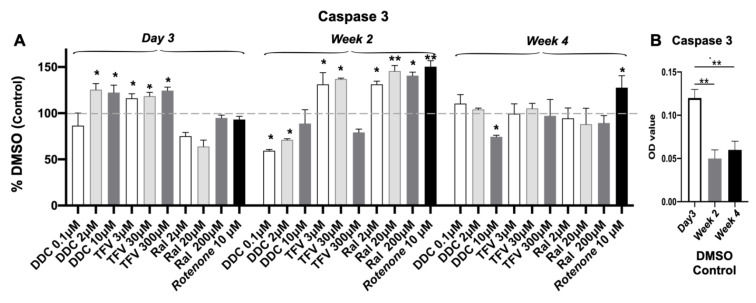
A significant increase in levels of Caspase-3 Levels in 3D USC spheroids after ART. (**A**). The levels of caspase 3 significantly increased at 1- and 2-weeks following treatment with ddC, TFV and RAL, particularly ddC at middle and high doses at day 3, TFV at the early and middle time points (day 3 and week 2), but RAL at all three doses at the middle time point (week 2) only. Rotenone induced significant higher levels of caspase 3 at middle and late time points (weeks 2 and 4). The cell lysates of USC were assessed by respective colorimetric assays. All were compared to DMSO (0.1%) control. (**B**). Levels of caspase-3 significantly decreased at DMSO control from day 3 to week 2 and 4. * *p* < 0.05; ** *p* < 0.01.

**Figure 7 pharmaceutics-14-01042-f007:**
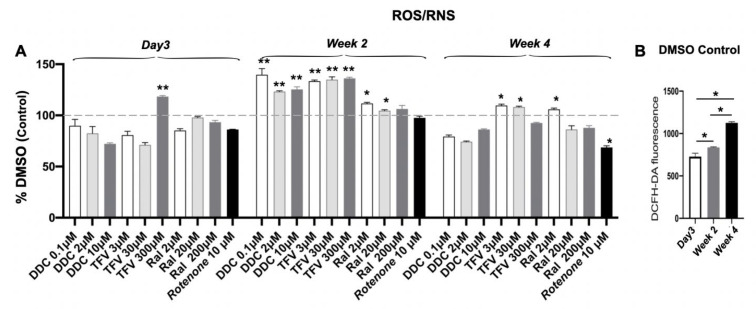
Expression of ROS/RNS in 3D USC spheroids. (**A**) significant increase in ROS/RNS levels was observed in 3D USC spheroids treated with ddC and TFV, compared to that in DMSO control at 2 weeks (*p* < 0.05). (**B**). Levels of ROS/RNS significantly increased with time at DMSO control during 4-week culture. * *p* < 0.05; ** *p* < 0.01.

**Table 1 pharmaceutics-14-01042-t001:** Assessment of chronic mitochondrial toxicity in 3D USC spheroids at 4 weeks.

Culture Models 3D USC Spheroids vs. 2D Culture of USC	
Mit-Tox Positive control:	Canonical drug known to induce Mito-Tox: Rotenone (RTNN)
ART controls:	NRTI known to induce significant Mito-Tox: Zalcitabine (ddC) NRTI known to induce minimal Mito-Tox: Tenofovir (TFV)
Negative control:	Solvent: DMSO (0.1%)
New drug:	INSTI with unclear Mito-Tox [33,34]: Raltegravir (RAL)
Measurements:	Cell growth rate, cell viability, OXPHOS Complex I, III, and IV, total ATP, Mitochondrial DNA copy number/mass, Caspase 3 for apoptosis, ROS/RNS for oxidative stress, total GSSG for major intracellular antioxidant

Abbreviations: USC—human urine-derived stem cells; Mito-Tox-mitochondrial toxicity;.NRTI—Nucleoside reverse transcriptase inhibitors; INSTI—Integrase Strand Transfer Inhibitor; DMSO—Dimethyl sulfoxide; OXPHOS-mitochondrial oxidative phosphorylation;ATP—Adenosine 5′-triphosphate; DNA-deoxyribonucleic acid; ROS—Reactive oxygen species; RNS—Reactive nitrogen species; GSSG-Glutathione disulfide. Notes: Three doses for each drug; 3D spheroids and 2D culture of USC were assessed at day 3, weeks 2, and 4 after exposure to drugs, respectively.

**Table 2 pharmaceutics-14-01042-t002:** Summary of antiretroviral drugs induced mitochondrial toxicity.

Independent Measurements	DMSO	ddC	TFV	RAL	RTNN
	(Neg Ctr)	(NRTI)	(NRTI)	(INSTI)	(Pos Ctr)
Size of spheroids	Stable	Drugs did not induce any effect on spheroid sizes			Reduced spheroid size at wk 4 (*p* < 0.01)
Cell number	Same or similar cell numbers were maintained during the 4-week culture				Decreased cell number at wk 2 and 4 (*p* < 0.01)
Percentage cell viability	~99%	~80%	~95%	~95%	~60%
Complexes I, III, and IV	No effect	Time- or dose-dependent decrease Complexes I, III, and IV (*p* < 0.05, *p* < 0.01)	No significant effect	Transiently decrease in Complex IV At wk 2 (*p* < 0.05)	Time-dependent decrease Complex I (*p* < 0.01)
Total ATP	Stable	decrease at high dose (*p* < 0.05; *p* < 0.01)	decrease at high dose (*p* < 0.05; *p* < 0.01)	decrease at middle and high doses (*p* < 0.05; *p* < 0.01)	Time-dependent decrease (*p* < 0.01)
Mitochondria-DNA Content	Stable	Dose- and time-dependent Decrease (*p* < 0.01)	No negative effect	Slightly decreased (*p* < 0.05)	(Compensatory) increased (*p* < 0.01)
Mt-DNA mass	No effect	Significantly decrease (*p* < 0.01)	Normal	Slightly decreased (*p >* 0.05)	Slightly Increased (*p >* 0.05)
Caspase 3 Activity	Increase at day 3; stable at wk 2, 4 (*p* < 0.01)	Increase at middle and high doses at wk 2 (*p* < 0.05; *p* < 0.01)	Increase at low and middle doses at wk 2 (*p* < 0.05)	Increase at all doses at wk 2 (*p* < 0.05, *p* < 0.01)	Increase at wk 2 and 4 (*p* < 0.05; *p* < 0.01)
ROS/RNS levels	Increase at wk 4 (*p* < 0.05)	Increase at wk 2 (*p* < 0.01)	Increase at wk 2 (*p* < 0.01)	Slightly increased (*p >* 0.05)	Decreased at wk 4 (*p* < 0.05)
Total GSSG	Stable	Drugs had no effect on total GSSG			

Abbreviations: Neg-Ctr—negative control; Pos-Ctr: Positive control; CCK-8: Cell Counting Kit-8; ATP—Adenosine triphosphate; NRTI: Nucleoside reverse transcriptase inhibitors; INSTI: integrase strand transfer inhibitor.

**Table 3 pharmaceutics-14-01042-t003:** Primers used in q-PCR analysis.

mtDNA Content (or mtDNA Copy Number)		
Hu Mt	Forward primer	CACCCAAGAACAGGGTTTGT
	Reverse primer	TGGCCATGGGTATGTTGTTA
Hu Nu	Forward primer	TGCTGTCTCCATGTTTGATGTATCT
	Reverse primer	TCTCTGCTCCCCACCTCTAAGT

All Primers were purchased from Integrated DNA Technologies, Coralville, IA.

**Table 4 pharmaceutics-14-01042-t004:** Comparison of mitochondrial toxicity assessment between 3D spheroids and 2D culture of human urine-derived stem cells.

Independent Measures	3D Spheroids	2D Culture
Cell growth	Remain consistent cell number over 4 weeks	Rapid, continuing proliferation and detaching after cell reach over-confluence.
Doubling time	No sign for cell proliferation	20 h at *p*4
Population doubling	The number of cells remained nearly the same during the 4-week culture	40–64 [19,20]
Cell viability/ cytotoxicity Assays	Both alive and dead cells can be identified by combined Mito-Tox with live/death kit	- Excessive cytoplasmic vacuolation after ddC treatment - Sensitively detected but only alive cells are accounted by MITO-TOX, CCK-8 and total ATP - Live/Dead kit might cause errors as dead cells were washed away
Mitochondrial function	- Cells grew slowly, with no necrosis at the center of spheroid at the size ≤ 350 µm at 4 weeks	- Cell proliferated fast and most energy from glycolysis, instead of OXPHOS-“Crabtree effect” - Not suitable for Mito-Tox
DNA replication	Slow with no cell proliferation	Rapid cell proliferation

Abbreviations: D: dimension; p-passage; Mito-Tox: mitochondrial toxicity; ATP—Adenosine triphosphate; OXPHOS-Oxidative phosphorylation.

## Data Availability

The data that support the finding of this study are available from the corresponding author upon reasonable request.

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
