# Peer review of "3D Spheroids of Human Primary Urine-Derived Stem Cells in the Assessment of Drug-Induced Mitochondrial Toxicity"

_pharmaceutics, 2022, doi:10.3390/pharmaceutics14051042_

Round 1
Reviewer 1 Report
This manuscript details the use of 3D spheroids generated from urine-derived stem cells as a method of identifying mitochondrial toxicity. The findings from this manuscript are novel and an important contribution to the field as they provide a preliminary platform of a longer-term in vitro method to assess mitochondrial toxicity. The manuscript is limited by small sample size, inconsistent result reporting, and cumbersome techniques, but the authors do acknowledge future studies can opt to utilize higher-throughput methods and gold-standard measures of mitochondrial health such as Seahorse extracellular flux assays.
The authors convincingly convey the need for in vitro assays to test and detect delayed drug-associated mitochondrial toxicity that only become evident after a longer exposure than what is usually done in existing in vitro toxicity models. However, there are several issues that should be addressed.
Major Comments
- Choice of antiretrovirals. Choosing zalcitabine (ddC) is an extreme choice and would amount to a second positive control. This drug is probably the most toxic antiretroviral ever used in humans and was quickly taken off the market as alternative agents were developed. Even over a few days of exposure, almost any cultured 2D cell model would reveal severe mitochondrial toxicity by ddC in the form of mtDNA depletion. A mid-range toxicity agent would have been much more relevant. Similarly, the choice of RAL is puzzling since of all INSTIs, it is the one for which no evidence of toxicity has been reported. Why not choose dolutegravir for which unexplained weight gain has been reported suggesting a potential metabolic/mitochondria involvement?
- Sample size and selection issues. The methods state that USCs were obtained from 8 healthy individuals, yet only three biological replicates were presented. The authors must state any specimen selection criteria applied or reasoning as to why only three were used (i.e. the other USCs were not able to be successfully propagated as a 3D spheroid, etc). Additionally, the authors should indicate the number of biological/independent replicates included in each figure. If only three replicates were done, statistics should not be employed.
- 2D control. Although passaging of 2D culture may introduce more variability, it is preferred over letting cells reach confluency as overconfluent cells will not provide reproducible results. It is hardly surprising that cells would detach altogether if let to become grossly over confluent. A more appropriate control would be 2D cultured cells passaged every time cells reached ~80% confluence (likely the 3-4-day mark) until the end of the 4-week period. The authors should comment on their choice to not passage or include follow-up experiments in which an additional 2D control with passage is used.
- It is unclear how ROS/RNS results were quantified. Were the fluorescence values normalized to the number of cells in the well? This would clearly influence the fluorescence output. This should be clarified. The same issue arises with the caspase measurement. The reported OD means little unless it is normalized to the number of cells in the well.
- Table 3. The Independent measurement in the first column do not need to repeat the techniques used; those were defined in the methods. What does “normal” mean?
- Figure 2. The 2D culture photographs do not allow the reader to appreciate changes. They are too small and show insufficient contrast.
- Errors between written results and figures. There are many instances in which the written results do not agree with the presented figures. For example, the written RAL results in section 3.5 state that “RAL at the middle dose of 20 μM significantly reduced mtDNA content (p<0.05) at 2 weeks”, but the figure does not show this and instead shows that RAL at the lowest dose of 2 μM increased mtDNA content (p<0.01) at 2 weeks. Another example includes the authors stating that “ddC reduced mtDNA content and mitochondrial mass at day 3, week 2, and week 4” yet they did not present mtDNA content results at day 3 and only showed mitochondrial mass images from week 4. There are no reasons to not include all results, in supplement if need be. The authors must ensure that the result section text is consistent with the data shown in the figures.
- Figure 4. The authors seem to selectively report the data. For example, attention is paid when ATP levels decrease but nothing is mentioned when they significantly increase as in Fig 4A, ddC day 3. The same selective reporting takes place for TFV increasing mtDNA content in Fig 5A.
- Contradictory statements on mtDNA content. In both the introduction and the discussion, the authors state that the 3D culture platforms have stable mitochondrial copy numbers. This is contradicted by their DMSO control results in which mtDNA content significantly increased (p<0.01) from week 2 to week 4. Additionally, the authors provide no explanation as to why mtDNA content from day 3 was not included. Either the data or an explanation should be provided.
- Throughout the manuscript, the authors report different concentrations for the high, medium, and low dose of ddC. Figures 2, 3, 4, 6, 7, and Supplemental Figure 1 report ddC concentrations of 0.1, 2, and 10 μM. Figure 5 reports 0.1, 2, and 20 μM. The abstract and discussion report ddC concentrations as 3, 30, and 300 μM. As the therapeutically relevant Cmax of ddC is ~0.1 μM, this inconsistency must be corrected.
- Although a 4-week assay may be superior to shorter toxicity assays, it is still unlikely to reveal long-term toxicity that develops following years of antiretroviral therapy. The authors should therefore tone down their language in the first paragraph of the discussion and conclusion. Generally, the authors tend to over-interpret the results presented which does not necessarily play in their favor and deters from the advantages their 3D model bring over existing assays.
Minor comments:
- The graphical abstract is too cluttered. The numerous bar graphs should be replaced by a diagram highlighting the main takeaways of the manuscript.
- Page 4. Given that short-term 2D cultures do detect some toxicity, the authors should tone down the language “Thus, short-term 2D cultures (≤ 2 weeks) of cell lines cannot ALWAYS detect chronic toxicity occurring in clinical settings, OR predict the long-term Mito-Tox relevance of drugs.”
- Two different plates were used for 2D cultures. It is unclear why and what effect this may have had on the results.
- Table 3. The Independent measurement in the first column do not need to repeat the techniques used; those were defined in the methods.
- The methods section is extraordinarily detailed and could benefit from a reduction by lumping similar sections together (i.e. the numerous ELISAs & colorimetric assays).
- The tables included to summarize the results are useful in presenting the data together but should include more details such as effect sizes and p-values. Statements such as “slightly decreased” have little scientific meaning.
- The authors should refrain using language such as “3.4 Total ATP content decreased after ART”. These experiments report the effect of exposure to a single antiretroviral; agent. This does not equate antiretroviral therapy which would involved multiple agent taken by a human.
- The abbreviation for hour or hours should always be “h”
- Page 5. More details should be provided on how “The spheroids were treated with an Ultrasonic Cell Disrupter System to help completely dissolve the crystal in the cells.” As this may not be sufficient information to allow one to repeat the work.
- PBS is defined twice
- Page 5. The described RT-PCR conditions seem wrong. “The PCR temperature cycling condition: initially denatured at 50 °C for 2 min, 95 °C for 10 min, followed by 40 cycles of denaturing at 95 °C for 30 seconds, then annealed at 60 °C for 1 second and extended at 95 °C for 15 seconds, 60°C for 1min, 95°C for 1 second.” The text suggests denaturation at 50 °C and extension at 95°C which makes no sense. Please verify. Also, sometimes it is RT-PCR, other times q-PCR, other times q-RT-PCR. This should be more consistent.
- Capitalize ELISA
- Inconsistent labeling (e.g. DDC vs ddC)
- Fig 5B. Typo. DMSO not DMSC
- What statistical software was used?
Author Response
We highly appreciate your insightful and constructive comments regarding the previous manuscript. We are encouraged by the positive comments. In this revised application, we have addressed the comments and revised the manuscript. Please find the revised paper in the attachment.
Major Comments
- Choice of antiretrovirals. Choosing zalcitabine (ddC) is an extreme choice and would amount to a second positive control. This drug is probably the most toxic antiretroviral ever used in humans and was quickly taken off the market as alternative agents were developed. Even over a few days of exposure, almost any cultured 2D cell model would reveal severe mitochondrial toxicity by ddC in the form of mtDNA depletion. A mid-range toxicity agent would have been much more relevant. Similarly, the choice of RAL is puzzling since of all INSTIs, it is the one for which no evidence of toxicity has been reported. Why not choose dolutegravir for which unexplained weight gain has been reported suggesting a potential metabolic/mitochondria involvement?
Reply: As pointed out, zalcitabine (ddC) selected as the second positive control in 3D culture is due to its significant mitotoxicity in 2D culture listed in the revised Table 1. Our data showed that ddC mediated dose- and time-dependent late MtT was detected in 4-wk culture 3D spheroid model, which cannot be detected in 2D cultures due to unstable cell growth.
As an integrase strand transfer inhibitor (INSTI), raltegravir (RAL) is controversial in its ability to cause Mito-Tox1, 2. Previous studies showed that dolutegravir and elvitegravir (EVG), but not raltegravir (RAL), showed significantly decreased cellular respiration in a dose-dependent manner3. However, recent studies showed that raltegravir (RAL) and dolutegravir directly impacted adipocytes and adipose tissue by inducing oxidative stress, mitochondrial dysfunction, fibrosis, and insulin resistance4. Thus, RAL was selected as a drug for testing Mio-Tox due to its unclear toxic effect. In our ongoing studies, more ART drugs (such as dolutegravir, bictegravir, and elvitegravir as INSTIs, islatravir as a nucleoside reverse transcriptase translocation inhibitor, and darunavir as a protease inhibitor) will be evaluated in this 3D system. We added this content in the Discussion section of the revised text.
- Sample size and selection issues. The methods state that USCs were obtained from 8 healthy individuals, yet only three biological replicates were presented. The authors must state any specimen selection criteria applied or reasoning as to why only three were used (i.e. the other USCs were not able to be successfully propagated as a 3D spheroid, etc). Additionally, the authors should indicate the number of biological/independent replicates included in each figure. If only three replicates were done, statistics should not be employed.
Reply: As suggested, we illustrated the inclusion criteria for the samples in section 2.2 .
“The urine samples from three individuals (at least three urine samples per donor) were mainly used as 3 biological independent replicates for all of tests in this study. Urine samples from other 5 individuals were used as the backup in this study.”
- 2D control. Although passaging of 2D culture may introduce more variability, it is preferred over letting cells reach confluency as overconfluent cells will not provide reproducible results. It is hardly surprising that cells would detach altogether if let to become grossly over confluent. A more appropriate control would be 2D cultured cells passaged every time cells reached ~80% confluence (likely the 3-4-day mark) until the end of the 4-week period. The authors should comment on their choice to not passage or include follow-up experiments in which an additional 2D control with passage is used.
Reply: As suggested, we have discussed passaging vs. non-passaging 2D USC culture in Mito-Tox testing in the discussion section in the revised text.
“ USC reaching over-confluence without passaging led to cell detachment when cells were exposed to the toxic drug for 4 weeks. An alternative approach could be to subculture and passage cells before they reach confluence and determine the cytotoxicity and Mito-Tox in 2D culture during 4-week drug toxicity testing. In this way, mtDNA copy number might increase with cell passages, which might affect the accuracy of the resulting Mito-Tox assessment. Further experiments could help to clarify this concern”.
- It is unclear how ROS/RNS results were quantified. Were the fluorescence values normalized to the number of cells in the well? This would clearly influence the fluorescence output. This should be clarified. The same issue arises with the caspase measurement. The reported OD means little unless it is normalized to the number of cells in the well.
Reply: For both the ROS/RNS assay and Caspase 3 assay, we used cell lysates, rather than cells. We measured the total protein content of the cell lysates using BCA assay. Based on measured concentration, we loaded the same concentration of cell lysate in each assay well as per assay protocol. The OD means were normalized to total protein of the cell lysates.
This statement was added at section 2.9
- Table 3. The Independent measurement in the first column do not need to repeat the techniques used; those were defined in the methods. What does “normal” mean?
Reply: As advised, we revised Table 3.
- Figure 2. The 2D culture photographs do not allow the reader to appreciate changes. They are too small and show insufficient contrast.
Reply: As suggested, we modified Figure 2
- Errors between written results and figures. There are many instances in which the written results do not agree with the presented figures.
For example, the written RAL results in section 3.5 state that “RAL at the middle dose of 20 μM significantly reduced mtDNA content (p<0.05) at 2 weeks”, but the figure does not show this and instead shows that RAL at the lowest dose of 2 μM increased mtDNA content (p<0.01) at 2 weeks.
Reply: Change was made as suggested.
“The new agent RAL at the low dose of 2 µM significantly reduced mtDNA content (p<0.05) at 2 weeks”.
Reply: We corrected the distribution according to what the graph shows.
Another example includes the authors stating that “ddC reduced mtDNA content and mitochondrial mass at day 3, week 2, and week 4” yet they did not present mtDNA content results at day 3 and only showed mitochondrial mass images from week 4. There are no reasons to not include all results, in supplement if need be. The authors must ensure that the result section text is consistent with the data shown in the figures.
Reply: We made the change as suggested.
“ddC decreasedmtDNA content at week 2 and week 4, and reduced mitochondrial mass at week 4”
- Figure 4. The authors seem to selectively report the data. For example, attention is paid when ATP levels decrease but nothing is mentioned when they significantly increase as in Fig 4A, ddC day 3. The same selective reporting takes place for TFV increasing mtDNA content in Fig 5A.
Reply: As suggested, we made changes in Fig 4A and Fig 5A.
“Significant increases in total ATP occurred 3 days after ddC treatment;”
“TFV at 30 µM and RAL at µM significantly increased mtDNA content (p<0.05) at 2 weeks.”
- Contradictory statements on mtDNA content. In both the introduction and the discussion, the authors state that the 3D culture platforms have stable mitochondrial copy numbers. This is contradicted by their DMSO control results in which mtDNA content significantly increased (p<0.01) from week 2 to week 4. Additionally, the authors provide no explanation as to why mtDNA content from day 3 was not included. Either the data or an explanation should be provided.
Reply: We rephrased this sentence: “With extended times for toxicity examination (shelf-life), mtDNA content increased 3-fold when the number of USC remained stable in 3D culture from weeks 2 to 4. In vitro 3D culture platforms seem to be better in testing late Mito-Tox under no cell proliferation with limited mtDNA replication conditions”.
The mtDNA content in the early stage of 3D culture (Day 3) was not tested as we focused on mtDNA content during the periods time when 3D spheroids maturely formed (i.e., week 2-4)
- Throughout the manuscript, the authors report different concentrations for the high, medium, and low dose of ddC. Figures 2, 3, 4, 6, 7, and Supplemental Figure 1 report ddC concentrations of 0.1, 2, and 10 μM. Figure 5 reports 0.1, 2, and 20 μM. The abstract and discussion report ddC concentrations as 3, 30, and 300 μM. As the therapeutically relevant Cmax of ddC is ~0.1 μM, this inconsistency must be corrected.
Reply: We corrected three doses of ddC (0.1, 2, and 10 μM) throughout the whole manuscript.
- Although a 4-week assay may be superior to shorter toxicity assays, it is still unlikely to reveal long-term toxicity that develops following years of antiretroviral therapy. The authors should therefore tone down their language in the first paragraph of the discussion and conclusion.
Reply: We agreed with the reviewer. 3D spheroids have limitation in Mito-Tox testing as patients are exposed to years of ART vs USC from collected from healthy donors only have weeks of drug exposure. However, USC from HIV human subjects or individuals after drug treatment or pre-exposure prophylaxis could be used as a cell source for establishing personalized toxicological assays to test ART-induced delayed Mito-Tox. We predict that use of HIV patients’ USC in 3D culture might better reflect chronic MitoTox of years of ART.
- Generally, the authors tend to over-interpret the results presented which does not necessarily play in their favor and deters from the advantages their 3D model bring over existing assays.
We agree with the reviewer, however, we want to err cautiously vs overextending the significance and especially the clinical applicability of our model in its early phase of development. This is the first of several planned manuscripts on this topic and our goal is to build a positive solid (albeit perhaps more conservative at first) groundwork for future studies. Future studies will also directly compare Mito-Tox in our 3D system to systemically administered ART both known and unknown Mito-Tox in animals and as mentioned above future studies will use USC from patients on long-term ART.
“The 3D USC spheroids are designed to be intermediate assays in toxicity assessment for drug development, to bridge the gap between traditional 2D cell culture and in vivo animal models, rather than replacing animal models of toxicity testing.” This sentence was added in the first paragraph.
Minor comments:
- The graphical abstract is too cluttered. The numerous bar graphs should be replaced by a diagram highlighting the main takeaways of the manuscript.
Reply: The graphical abstract was revised as suggested
- Page 4. Given that short-term 2D cultures do detect some toxicity, the authors should tone down the language “Thus, short-term 2D cultures (≤ 2 weeks) of cell lines cannot ALWAYS detect chronic toxicity occurring in clinical settings, OR predict the long-term Mito-Tox relevance of drugs.”
Reply: We made the change as suggested.
- Two different plates were used for 2D cultures. It is unclear why and what effect this may have had on the results.
Reply: The reviewer might ask about “two different plates were used for 3D cultures”.
We used two types of 3D spheroids for different purposes: 96 plates were used mainly used for daily observing the morphology of spheroids and for monitoring cell viability; the 81-well plates are designed for use in cytotoxicity, western-blot analysis and ELISA as can generate more cells. The spheroids generated from both types of 3D culture devices produced similar results. We revised the sentences in Materials and Methods section 2.3 as listed below:
To establish 3D spheroid cultures, two types of ultralow attachment (ULA) well plates were used: a) ULA 96 well plates (Corning, Glendale, Arizona) for USC plated in 200 µl culture medium with each well having 8x103 cells to observe morphology of spheroids, measure cell growth and viability, mt-DNA content by quantitative real time-PCR (q-PCR), and mitochondrial mass by immunofluorescence staining, with the media changed every other day; b) 81 spheroids (8x103 cells/spheroid/well) were placed in a “81-well micro-mold plate” (MicroTissues 3D Petri Dish, Sigma, St. Louis, MO); six “81-well micro-mold plates” were placed in a ULA 6-well-plate (Sigma), which was convenient for daily changing medium. This “81-well micro-mold” was mainly used in testing cytotoxicity and Mito-Tox with enzyme-linked immunosorbent assay (ELISA) as this micro-mold is designed to generate sufficient cell numbers. Total ATP content, Caspse-3 and total GSSG with Colorimetric assays, and ROS/RNS with fluorescence assay (Tables 1 and 3). Drugs were added into the medium on the third day after seeding.
- Table 3. The Independent measurement in the first column do not need to repeat the techniques used; those were defined in the methods.
Reply: As advised, the techniques in the first column are removed.
- The methods section is extraordinarily detailed and could benefit from a reduction by lumping similar sections together (i.e. the numerous ELISAs & colorimetric assays).
Reply: The Methods of the ELISA part were simplified as suggested.
- The tables included to summarize the results are useful in presenting the data together but should include more details such as effect sizes and p-values. Statements such as “slightly decreased” have little scientific meaning.
Reply: As suggested, we revised the tables.
- The authors should refrain using language such as “3.4 Total ATP content decreased after ART”. These experiments report the effect of exposure to a single antiretroviral; agent. This does not equate antiretroviral therapy which would be involved multiple agent taken by a human.
Reply: ART infers MULTIPLE agents, we used just one at a time. So instead of ART just list that one drug. We made these changes to the text in the manuscript
The abbreviation for hour or hours should always be “h”
Reply: Change was made as suggested.
- Page 5. More details should be provided on how “The spheroids were treated with an Ultrasonic Cell Disrupter System to help completely dissolve the crystal in the cells.” As this may not be sufficient information to allow one to repeat the work.
Reply: More details were given about how the Ultrasonic Cell Disrupter System works as suggested.
- PBS is defined twice
Reply: The second definition of PBS was deleted.
- Page 5. The described RT-PCR conditions seem wrong. “The PCR temperature cycling condition: initially denatured at 50 °C for 2 min, 95 °C for 10 min, followed by 40 cycles of denaturing at 95 °C for 30 seconds, then annealed at 60 °C for 1 second and extended at 95 °C for 15 seconds, 60°C for 1min, 95°C for 1 second.” The text suggests denaturation at 50 °C and extension at 95°C which makes no sense. Please verify. Also, sometimes it is RT-PCR, other times q-PCR, other times q-RT-PCR. This should be more consistent.
Reply: We checked the process and corrected the description of the q-PCR methods as suggested.
- Capitalize ELISA
Reply: All “ELISA”s were capitalized.
- Inconsistent labeling (e.g. DDC vs ddC)
Reply: All “DDC”s were charged to “ddC”.
- Fig 5B. Typo. DMSO not DMSC
Reply: Correction was done as suggested.
- What statistical software was used?
Reply: GraphPad Prism version 9.0.0 (GraphPad Software, San Diego, California USA) was used for statistical analyses. This information was added in the methods part.
[1] Brehm TT, Franz M, Hufner A, Hertling S, Schmiedel S, Degen O, Kreuels B, Schulze Zur Wiesch J: Safety and efficacy of elvitegravir, dolutegravir, and raltegravir in a real-world cohort of treatment-naive and -experienced patients. Medicine (Baltimore) 2019, 98:e16721.
[2] Gorwood J, Bourgeois C, Pourcher V, Pourcher G, Charlotte F, Mantecon M, Rose C, Morichon R, Atlan M, Le Grand R, Desjardins D, Katlama C, Feve B, Lambotte O, Capeau J, Bereziat V, Lagathu C: The Integrase Inhibitors Dolutegravir and Raltegravir Exert Proadipogenic and Profibrotic Effects and Induce Insulin Resistance in Human/Simian Adipose Tissue and Human Adipocytes. Clin Infect Dis 2020, 71:e549-e60.
[3] Korencak M, Byrne M, Richter E, Schultz BT, Juszczak P, Ake JA, Ganesan A, Okulicz JF, Robb ML, de Los Reyes B: Effect of HIV infection and antiretroviral therapy on immune cellular functions. JCI insight 2019, 4.
[4] Gorwood J, Bourgeois C, Pourcher V, Pourcher G, Charlotte F, Mantecon M, Rose C, Morichon R, Atlan M, Le Grand R, Desjardins D, Katlama C, Fève B, Lambotte O, Capeau J, Béréziat V, Lagathu C: The Integrase Inhibitors Dolutegravir and Raltegravir Exert Proadipogenic and Profibrotic Effects and Induce Insulin Resistance in Human/Simian Adipose Tissue and Human Adipocytes. Clinical Infectious Diseases 2020, 71:e549-e60.

Reviewer 2 Report
The manuscript titled "3D Spheroids of Human Primary Urine-Derived Stem Cells in the Assessment of Mitochondrial Toxicity Induced by Antiretroviral Therapy" is a very interesting scientific article on the us of spheroids for the evaluation of mithocontrial damages of antiviral therapies.
Mnauscript is well written, methods are clear and results are well described. References are of good quality, however, some improvements are needed:
1)Please, improve the introduction and discussion with a proper description of new pharmacological and non pharmacaological approach to reduce mithocondrial damages of antiviral therapies, authors should explain how these methods should studied in spheroids models and the role of CD44 targeting in this direction. Authors should discuss on the role of hyaluronic acid hydrogels as natural CD44 targeting able to improve mithcontrial functions ( cite 10.1007/s10856-013-4895-4 )
2) please describe the possible study of antidiabetic drugs in mithocondrial protection during antiviral therapies also through shperoids models.
Author Response
We highly appreciate your insightful and constructive comments regarding the previous manuscript. We are encouraged by the positive comments. In this revised application, we have addressed the comments and revised the manuscript. Please find the revised paper in the attachment.
Major Comments
The manuscript titled "3D Spheroids of Human Primary Urine-Derived Stem Cells in the Assessment of Mitochondrial Toxicity Induced by Antiretroviral Therapy" is a very interesting scientific article on USC of spheroids for the evaluation of mitochondrial damages of antiviral therapies.
The manuscript is well written, the methods are clear, and the results are well described. References are of good quality; however, some improvements are needed:
- Please, improve the introduction and discussion with a proper description of the new pharmacological and nonpharmacological approach to reducing mitochondrial damage of antiviral therapies, authors should explain how these methods should be studied in spheroids models and the role of CD44 targeting in this direction. Authors should discuss the role of hyaluronic acid hydrogels as natural CD44 targeting able to improve mitochondrial functions (cite 10.1007/s10856-013-4895-4)
Reply: As advised, we have discussed how to improve long-term cell viability in 3D culture for assessing chronic Mito-Tox or reducing mitochondrial damage of antiviral therapies with new pharmacological and nonpharmacological approaches, such as using hyaluronic acid hydrogels as natural CD44, or silk-fiber matrix as we are currently developing. We will determine the role of hyaluronic acid hydrogels as natural CD44 targeting able to improve mitochondrial functions in future studies.
We added this content and cited the article in the section of Discussion as the reviewer suggested. “Recent studies showed new pharmacological and nonpharmacological approaches, such as hyaluronic acid hydrogels as natural CD44 targeting gels could improve long term cell retention of mitochondrial function in vitro. [PMID: 23471500]. It would be beneficial to use hyaluronic acid hydrogels or silk fiber matrices as we are developing to improve long-term cell viability in 3D culture for assessing chronic toxicities or reducing mitochondrial damage of antiviral therapies in future studies. Thus, a 3D culture system with new biomaterials might carry more cells stably at the level of the mitochondria for longer-term culture for Mito-Tox assessment.”
- 2)Please describe the possible study of antidiabetic drugs in mitochondrial protection during antiviral therapies also through spheroids models.
“Anti-diabetic drugs, such as Miglitol, can inhibit oxidative stress-induced apoptosis and mitochondrial ROS over-production in endothelial cells by enhancing AMP-activated protein kinase [PMID: 23018899]. Thus, our 3D spheroid model might be used in testing mitochondrial anti-diabetic drugs in prevention of mitochondrial side effects of ART and other agents.”
This content was added in the discussion section of the manuscript.

Round 2
Reviewer 1 Report
The authors adequately addressed the concerns and the manuscript is improved.